# Effect of Lipophilic Chains on the Antitumor Effect of a Dendritic Nano Drug Delivery System

**DOI:** 10.3390/molecules28010069

**Published:** 2022-12-21

**Authors:** Lijuan Ding, Xiangtao Wang, Ting Wang, Bo Yu, Meihua Han, Yifei Guo

**Affiliations:** 1Institute of Medicinal Plant Development, Chinese Academy of Medical Sciences & Peking Union Medical College, No. 151 Malianwa North Road, Haidian District, Beijing 100193, China; 2Key Laboratory of Bioactive Substances and Resources Utilization of Chinese Herbal Medicine, Ministry of Education, Chinese Academy of Medical Sciences & Peking Union Medical College, Beijing 100093, China; 3Beijing Key Laboratory of Innovative Drug Discovery of Traditional Chinese Medicine (Natural Medicine) and Translational Medicine, No. 151 Malianwa North Road, Haidian District, Beijing 100193, China

**Keywords:** aliphatic chain, amphiphilic nanocarriers, hybrid structure, morphology, antitumor activity

## Abstract

Oligoethylene glycol dendron (G2) has been used in drug delivery due to its unique dendritic structure and excellent properties. In order to investigate the effects of lipophilic chains on drug delivery, the amphiphilic hybrid compound G2-C18 is synthesized, and celastrol (CSL) is selected to prepare “core-shell” structured CSL-G2-C18 nanoparticles (NPs) via the antisolvent precipitation method. Meanwhile, CSL-G2 NPs are prepared as the control. The two NPs show similar particle sizes and polydispersity indexes, while their morphologies exhibit dramatic differences. CSL-G2 NPs are solid spherical particles, while G2-C18 NPs are vesicles. The two NPs present ideal stability and similar release tendencies. The in vitro toxicity results show that the cell inhibition effect of CSL-loaded NPs is significantly enhanced when compared with free CSL, and the antitumor effect of CSL-G2-C18 NPs is stronger than that of CSL-G2 NPs. The IC_50_ value of CSL-G2 NPs and CSL-G2-C18 NPs is enhanced about 2.8-fold and 5-fold when compared with free CSL, respectively. The above results show that lipophilic chain-linking dendritic hybrid nanocarriers promote antitumor activity by affecting the morphology of NPs, which may aid in the selection of carrier designs.

## 1. Introduction

Nano drug delivery systems (NDDS) involve the use of different nanocarriers to deliver drugs to specific tissues/organs, improve drug stability, reduce drug toxicity, and prolong plasma half-life [1]. Different nano delivery systems, e.g., liposomes [2], micelles [3], nanoparticles (NPs) [4], nanosuspensions [5], vesicles [6], etc., have been developed. The selection of a suitable carrier is crucial in the construction of a NDDS. Amphiphilic copolymers present good water solubility due to their hydrophilic block and a good drug-loading capacity due to the interactions between the hydrophobic block and drugs, meaning that they have been studied as nanocarriers [7,8,9,10,11]. Amphiphilic copolymers can self-assemble with hydrophobic drugs into core–shell structures. The hydrophobic drug is loaded into the hydrophobic core through different intermolecular forces such as hydrophobic interactions, hydrogen bonding, van der Waals forces, and so on [12,13]. The hydrophilic segments distributed on the carrier surface provide a hydrophilic shell for the NDDS. This core–shell structure can effectively reduce protein adsorption, evade the recognition of the mononuclear phagocytic system, and extend the circulation time of the drug [12,14].

To improve the antitumor efficacy of NDDS, the components of nanocarriers should be optimized in detail. Polyethylene (PEG), which has good aqueous solubility and biosafety is utilized broadly as a hydrophilic block to form amphiphilic copolymers and construct amphiphilic nanocarriers. By reviewing the literature [15,16], it can be found that the phospholipid bilayer membrane has a good embedding effect on aliphatic chains, which can improve the affinity of NPs to the cell membrane, promote the fusion of cell membranes, and accelerate the entry of NPs into cells through endocytosis. Jana et al. also found that multivalent interactions between fatty chains and intracellular biomolecules can enhance the level of reactive oxygen species in cells and induce apoptosis, which is beneficial for tumor therapy [17]. In addition, the hydrophobic interaction between the aliphatic chain and the hydrophobic drug is utilized to prolong the drug release [18]. Therefore, in our previous study, PEG was conjugated with octadecylamine to prepare an amphiphilic compound. We then encapsulated hydroxycamptothecin to construct a NDDS with high anticancer activity [19].

In addition, it has been reported that antitumor activity is related to the morphology of a NDDS, which could be affected by the structure of nanocarriers. A drug delivery system with branched structures as the hydrophilic segment can involve rods [20,21], sheets [22,23], vesicle-shaped particles [24,25], etc., which exhibits better antitumor effects, while drug-loaded NPs with linear PEG as nanocarriers always have a spherical structure [26,27]. Branched oligoethylene glycol (OEG) compounds, as a homolog of PEG, exhibit similar long-circulation properties and good biocompatibility. Gillich et al. found that dendritic OEG NPs have smaller average particle sizes, better stability, and better aggregation reversibility when compared with linear analogs, which can be applied in drug delivery [28]. In our previous study, it was found that a NDDS based on a branched OEG dendron (G2) nanocarriers presented non-spherical morphology, and resulted in enhanced antitumor activity [29]. 

Based on these results of combining the advantages of G2 dendron and aliphatic chain C18, in this study, the amphiphilic compound G2-C18 was utilized as a nanocarrier to construct celastrol nanoparticles (CSL NPs), and the structural effect of the nanocarriers on the cytotoxicity of the NDDS was studied further. CSL NPs were prepared successfully via the antisolvent precipitation method, their drug-loading contents, particle sizes, morphologies, physical states of the CSL in the NPs, in vitro release profiles, and in vitro cytotoxicity were evaluated in detail.

## 2. Results and Discussion

### 2.1. Preparation and Characterization of Celastrol-Loaded Nanoparticles (CSL NPs)

The structures of two kinds of nanocarriers are shown in Figure 1, including a hydrophilic G2 and an amphiphilic hybrid compound (G2-C18). Amphiphilic dendritic nanocarriers G2-C18 was synthesized via the attach-to method, in which G2 was conjugated with an aliphatic C18 chain via an amide bond. Next, CSL drug-loaded NPs that could self-assemble into a “core–shell” structure in an aqueous solution were prepared via the antisolvent precipitation method (Figure 1). The mean particle size of CSL-G2 NPs was 135.6 ± 2.4 nm, and the polydispersity index (PDI) was 0.09 ± 0.001. After being detected by dynamic light scattering (DLS), the CSL-G2-C18 NPs had a similar particle size and PDI, it was 121.4 ± 5.2 nm, 0.11 ± 0.03. The two kinds of NPs had similar drug-loading content (DLC) after detection via HPLC, which were 85.8 ± 1.7% and 86.4 ± 2.4% for CSL-G2 NPs and CSL-G2-C18 NPs, respectively.

After detection via transmission electron microscope (TEM), CSL-G2 NPs showed a regular spherical morphology with a particle size of approximately 70 nm (Figure 2c); CSL-G2-C18 NPs presented a vesicular morphology with a particle size of 100 nm (Figure 2d). The particle sizes of the two NPs when detected by TEM were smaller than when using DLS. The variation in particle size can be explained by the different measurement processes. The particle size measured by DLS is the hydrodynamic diameter of the nanoparticles in an aqueous solution, while TEM detects the actual diameter of the dried nanoparticles.

### 2.2. Self-Assembly Behavior

Based on the TEM images, these two CSL NPs presented different morphologies, which revealed the different self-assembly processes. To study the self-assembly processes in detail, blank G2 NPs and G2-C18 NPs were detected under the same conditions. The results are shown in Figure 2. Blank G2 NPs presented a spherical morphology with a diameter of approximately 50 nm (Figure 2a), while G2-C18 NPs showed irregular spheres with a larger particle size than that of blank G2 NPs (Figure 2b). The different morphologies can be attributed to the different structures of G2 and G2-C18. The hydrophobic/hydrophilic volume ratio is the key parameter in amphiphilic copolymers, which significantly affects the self-assembly process and results in different morphologies. When the hydrophobic volume increases, the aggregates’ morphologies are spherical micelles, cylindrical micelles, and vesicles [29]. In G2′s structure, only the benzyl ester played a hydrophobic role, and thus the hydrophobic volume fraction of G2 was low, which induced G2 to form a spherical morphology. After conjugating with C18, the hydrophobic volume fraction of G2-C18 was increased significantly. Therefore, it seems that G2-C18 aggregated and formed irregular spheres, which looked like cylindrical micelles.

To further study the self-assembly property of G2 and G2-C18, the critical micellar concentration (CMC) was detected via pyrene fluorescence spectroscopy [30,31], the results are shown in Figure 3. It can be seen that when the concentration reaches 15.7 μg/mL, the intensity ratio increases sharply, which suggests that G2-C18 may self-assemble to form aggregates at a concentration of 15.7 μg/mL. A similar phenomenon is exhibited in a G2 solution, while the CMC of G2 is 121.4 μg/mL. The CMC decreases when increasing the hydrophobic portion of amphiphilic copolymers, which is consistent with previous reports [32,33,34].

After being utilized as nanocarriers to prepare nanoscale drug delivery system, G2 and G2-C18 entrap CSL via hydrophobic interactions to form stable CSL-loaded nanoparticles, but the assembly process is significantly different between these two nanocarriers. For G2, due to its small hydrophobic fraction, it was speculated that CSL forms the inner core of the nanoparticles in an amorphous state, then G2 disperses on the surface via hydrophobic interactions. Therefore, the CSL-G2 NPs were observed as bright solid balls in TEM images (Figure 2c). On the contrary, the hydrophobic portion of G2-C18 was enhanced significantly after conjugating the C18 chain, which promoted hydrophobic interactions between the nanocarrier and the drug. During the assembly process, CSL molecules interacted with C18 chains to form the hydrophobic portion of the nanoparticles, further enhancing the hydrophobic volume ratio. Hence, in contrast to the cylindrical morphology of blank G2-C18 nanoparticles, CSL-G2-C18 NPs presented a vesicular morphology (Figure 2d).

### 2.3. Differential Scanning Calorimetry (DSC) Analysis

DSC was utilized to analyze the physical state of CSL in these NPs (Figure 4). CSL presented a sharp endothermic peak at 151.3 °C and an exothermic peak at 212.8 °C, which were attributed to the melting point and the degradation of CSL, respectively. Similar peaks appeared in the thermogram of both physical mixtures. Only one exothermic peak was shown in each thermogram at 218.8 and 219.2 °C for CSL-G2 NPs and CSL-G2-C18 NPs, respectively. The characteristic endothermic peaks disappeared in these two NPs. Moreover, the exothermic peaks shifted to slightly higher temperatures. According to published reports [35,36], these results suggest that the hydrophobic interactions between the CSL and nanocarriers induced CSL to disperse in these two NPs as an amorphous form or a molecular dispersion.

### 2.4. X-ray Powder Diffraction (XRD) Analysis

To further clarify the interaction between CSL and nanocarriers, XRD was applied to illustrate the crystal characteristics of drug, drug-loaded NPs, and physical mixtures of drug and nanocarriers (Figure 5). The XRD patterns of CSL showed its characteristic peaks at 2θ ranging from 5 to 25°, especially at 6.3, 11.8, 13.4, 14.2, 14.6, 16.0, 16.9, and 22.9°. These sharp signal peaks indicated the highly crystalline structure of CSL, which was consistent with previous reports [37,38]. For the physical mixtures of G2/CSL and G2-C18/CSL, several intense peaks were shown in the diffractogram, including at 12.0, 13.4, 14.2, 14.6, 16.0, and 16.9°. These results suggested that the crystalline structure of CSL was maintained in these physical mixtures. However, there was no obvious diffraction peak in CSL-loaded NPs, and the characteristic peak of CSL also disappeared, demonstrating that during the encapsulation of the drug into the NPs, the crystalline CSL was transformed into an amorphous state, which suggests that the CSL interacted with the nanocarriers and was dispersed in the hydrophobic core. As expected, the XRD conclusion is consistent with the DSC results.

### 2.5. Storage Stability

To estimate their storage stabilities, the two CSL NPs were stored at 4 °C for 28 days, and particle size and PDI were recorded during the entire process (Figure 6). During the entire storage process, no turbidity or precipitation was observed. Both the particle size and PDI of these two CSL NPs were maintained well. The particle size was 137.6 ± 2.9 and 127.7 ± 5.1 nm for CSL-G2 NPs and CSL-G2-C18 NPs, respectively. These results suggested that these two CSL NPs presented good storage stability.

### 2.6. Media Stability Analysis

The media stability was measured to evaluate whether the CSL NPs are suitable to treat mice via intravenous administration (Figure 7). Intravenous administration requires NPs solutions to be stable in normal saline or glucose solution to avoid the hemolysis of red blood cells; hence, the stability of NPs was evaluated in glucose solution, and both CSL NPs presented good stability in glucose solution. The particle size was maintained at 150 and 200 nm for CSL-G2 NPs and CSL-G2-C18 NPs, respectively. After mixing with plasma, the particle size of CSL-G2 NPs was enlarged from 135.6 to 291.8 nm, and from 121.4 to 624.9 nm for CSL-G2-C18 NPs. Although the particle sizes changed within the initial 2 h, they were maintained well over the following 10 h; no obvious aggregation or sedimentation phenomena were observed during the entire experimental process. The size increase of CSL NPs in plasma may be attributed to interactions between the NPs and plasma, which are affected by the proteins, electrolytes, and ionic strength [39,40,41]. It seemed that both CSL NPs presented good stability in blood circulation. The blood circulation time of NPs in the human body is approximately 2–3 h; therefore, the media stability was evaluated for 12 h.

### 2.7. CSL-Loaded NPs Release Characteristic Analysis

In order to comparatively evaluate the release characteristics of CSL loaded with different nanocarriers, the cumulative release curves of the two NPs in 5% glucose solution were plotted. The results are shown in Figure 8. According to the release curves, the complete release of free CSL, CSL-G2 NPs, and CSL-G2-C18 NPs was achieved after 12 h, four days, and six days, respectively. Compared with CSL DMSO solution, these two CSL-loaded NPs showed sustained release behaviors in vitro. These results are consistent with published papers and may be explained by the structure of NPs [42,43]. Moreover, a similar release rate for these two NPs within the initial 12 h was observed, which was 27.9% for CSL-G2 NPs and 23.8% for CSL-G2-C18 NPs. However, the release rate of CSL-G2 NPs was higher over the subsequent three days. It has been reported that the in vitro release behavior of NPs is related to the length of hydrophobic chain of the carrier [44]. The lower release rate of CSL-G2-C18 may be due to the stronger hydrophobic interaction between C18 and CSL. 

### 2.8. In Vitro Cytotoxicity Analysis

The 4T1 cells were incubated with the two kinds of NPs over the concentration range 0.25–2.5 µg/mL (equivalent concentration of CSL), and the in vitro cytotoxicity of the CSL NPs was studied via an MTT method. Figure 9 shows the results of the cell inhibition rate. After incubation with 4T1 cells for 48 h, the IC_50_ values of free CSL, CSL-G2 NPs, and CSL-G2-C18 NPs were 1.55, 0.56, and 0.31 µg/mL, respectively. The in vitro proliferation inhibition rate of the two NPs was significantly higher than that of the free CSL (*p* < 0.05), and the IC_50_ values decreased approximately 2.8-fold and 5-fold with the CSL-G2 NPs and CSL-G2-C18 NPs, respectively. This phenomenon may be explained by different cellular uptake mechanisms. According to previous research, it is reasonable to speculate that free CSL molecules cross the cell membrane by passive transport, which makes it easier to pump out via p-glycoprotein-mediated multidrug resistance. NPs accumulate at the tumor site through the enhanced permeability and retention effect and are then taken up by tumor cells through the endocytic pathway, which increases the cell uptake efficiency of small molecule drugs such as CSL and improves the inhibitory effect on tumor cells. Furthermore, the cytotoxicity of CSL-G2-C18 NPs is stronger than CSL-G2 NPs, which might mean that the lipophilic segment of C18 can better bind to the phospholipid bilayer membrane during the process of internalized endocytosis, thereby enhancing the inhibitory effect on 4T1 cell proliferation. Furthermore, the vesicular structure of CSL-G2-C18 NPs is more flexible than the compact spherical structure of the CSL-G2 NPs, which may increase the endocytosis efficiency by changing its aspect ratio. 

## 3. Materials and Methods

### 3.1. Chemical Reagents and Cell Line

Dendritic oligoetheylene dendron (G2) and amphiphilic nanocarrier G2-C18 were synthesized according to previously described methods [29,45]. Celastrol (CSL, purity >98%) was obtained from Aktin Chemicals, Inc. (Chengdu, China). Dialysis membrane (MWCO = 14,000 Da) was purchased from Spectrum Laboratories Inc (Los Angeles, CA, USA). Acetonitrile, methanol, and phosphoric acid for chromatography were obtained from Fisher Scientific (Pittsburgh, PA, USA). Other analysis-level reagents and solvents were purchased and used without further purification. 

The murine-derived breast cancer (4T1) cell line was acquired from the Institute of Basic Medical Science, Chinese Academy of Medical Science (Beijing, China) and cultured in RPMI-1640 medium containing 10% fetal bovine serum and 100 units/mL penicillin G and streptomycin. The culture environment was maintained at 37 °C and 5% CO_2_ atmosphere.

### 3.2. Preparation of CSL-Loaded NPs

CSL-loaded NPs formed by self-assembly in aqueous solution were prepared via the ultrasonication-dialysis method. The specific steps were as follows: 8 mg CSL and 4 mg nanocarriers (G2 and G2-C18) were dissolved in 1 mL N,N-dimethylformamide (DMF) at 25 °C, then the organic phase was injected dropwise into 5 mL deionized water using an ultrasonic bath (Ultrasonic Cleaner, Kun Shan Ultrasonic Instruments Co., Ltd., Kun Shan, China) at 25 °C and 250 W. After sonication for 10 min, the mixture of organic and aqueous phases was subjected to a dialysis bag (MWCO = 14,000 Da). Deionized water (4 × 1 L) was used as the dialysis medium and was changed every one hour to remove DMF and unloaded CSL. The concentration of CSL in the prepared NPs was quantitatively analyzed by HPLC (Ultimate 3000, DIONEX, Sunnyvale, CA, USA) with a thermos C18 column (4.60 mm × 250 mm, 5 µm). According to the HPLC detection method, the calibration curve for the CSL concentration (Y = 0.6094 X − 0.0604, R^2^ = 0.9998) was established. Acetonitrile: water (phosphoric acid, 0.005 M) (85/15, *v/v*) was used as the mobile phase and was run at a flow rate of 1 mL/min. The sample injection volume was set to 20 μL, and a wavelength of UV 425 nm was used. Then, we calculated the drug-loading content (DLC) of the CSL-loaded NPs as follows: DLC% = (weight of loaded CSL/weight of CSL-loaded NPs) × 100%

### 3.3. Characterization of CSL-Loaded NPs

The dynamic light scattering (DLS) method was used to characterize the physicochemical properties of the CSL drug-loaded NPs. The average hydrodynamic diameter, polydispersity index (PDI), and zeta potential were detected by a DLS spectrophotometer (Zetasizer Nano ZS, Malvern Instruments, Malvern, UK) with the scattering angle θ = 173° at 25 °C. Each sample was diluted to about 1 mg/mL of the CSL equivalent concentration, and measurements were repeated three times.

### 3.4. Morphology of CSL-Loaded NPs by Transmission Electron Microscope (TEM)

The morphological characterization of the CSL NPs was observed under a TEM (JEM-1400, JEOL, Tokyo, Japan) with an acceleration voltage of 120 kV. Before observation, a negative dyeing method was used for the NPs. Briefly, 5 μL of the sample solution (100 μg/mL) was dripped onto a copper net (300 mesh). After air drying, an appropriate amount of 2% (*w*/*v*) uranyl acetate solution was added dropwise for 60 s for dyeing, and the excess staining solution was absorbed and dried.

### 3.5. Pyrene Fluorescence Spectroscopy

Pyrene was utilized as the fluorescence probe to detect the critical micellar concentration of G2 and G2-C18. Briefly, pyrene was dissolved in acetone and added into an Eppendorf tube. After evaporating the acetone, pyrene (6.0 × 10^−5^ mol) was left in each tube. G2 and G2-C18 solutions with concentrations ranging from 1 μg/mL to 1 mg/mL were added to the tubes separately. The mixtures were sonicated for 10 min and stirred at room temperature overnight. The fluorescence spectra were measured via an F-4500 FL Spectrophotometer at an excitation wavelength of 334 nm.

### 3.6. Differential Scanning Calorimetry Study (DSC)

A series of sample powders (lyophilized powder of CSL-loaded NPs, CSL and nanocarriers bulk powder, and the mixture of CSL and nanocarriers) were subjected to differential calorimetric analysis via a differential scanning calorimeter (DSC, Q200, TA Instruments, New Castle, DE, USA). A total of 5 mg of accurately weighed sample powder was placed in a standard aluminum pan and sealed in a nitrogen atmosphere. Then, the DSC thermogram was obtained by scanning each sample in the range of 0–400 °C at a scan speed of 10 °C/min.

### 3.7. X-ray Diffraction Study (XRD)

The X-ray crystallographic patterns of the three kinds of samples (lyophilized powder of CSL-loaded NPs, CSL bulk powder, and the mixed powder of CSL and nanocarriers) were investigated using an X-ray diffractometer (XRD, D8 advance, Bruker, Karlsruhe, Germany). The sample powder was placed on a quartz shelf and pressed into a circle. The detection parameters were set to 100 mA and 45 kV. We scanned from 3° to 80° of 2θ, with the step size of 0.01° at 3 s/step.

### 3.8. Storage Stability Study

The NPs solutions were stored at 4 °C for 28 days. The particle size and PDI were recorded at predetermined times. The experiment was carried out in triplicate.

### 3.9. Medium Stability Study

To evaluate and compare the medium stability, two CSL NPs were mixed with 10% glucose solution at a volume ratio of 1/1 and plasma at a volume ratio of 1/4, and were incubated at 37 °C. The particle size and PDI of the NPs were measured at 0, 2, 4, 6, 8, 10, and 12 h. The occurrence of precipitation and aggregation was observed at the same time. Each sample was detected in triplicate.

### 3.10. Investigation on the Release Profile of CSL-Loaded NPs In Vitro

The dialysis method was applied to plot the cumulative drug release profile of CSL-loaded NPs and compare the release kinetics of the NPs. In a typical process, the CSL-loaded NPs solution (2 mL, 5 mg/mL) was placed into a dialysis bag (MWCO = 14,000 Da), and immersed into the release medium, which contained 5% glucose solution (50 mL) at 37 °C under sink conditions at a speed of 150 rpm. At specific time intervals, 5 mL of release medium was withdrawn to measure the concentration of released CSL via UV-HPLC at 425 nm, and an equal amount of fresh medium was replenished at the same time. The release medium was changed every 24 h. The samples were measured three times to calculate the cumulative release and draw the drug cumulative release profile.

### 3.11. In Vitro Cytotoxicity Assay

The cell viability of 4T1 cells was measured via the MTT method to evaluate the antitumor effect of the CSL NPs in vitro. The 4T1 cells in the logarithmic growth phase were inoculated in a 96-well plate (8.0 × 10^3^ cells/well) at 37 °C and 5% CO_2_ for 24 h, then the medium was replaced with fresh RPMI-1640 and the CSL-loaded NPs were diluted to different concentrations of 0.25, 0.5, 0.75, 1, 1.5, 2, 2.5 μg/mL and added to each well. After 48 h of incubation, 20 μL of MTT solution (5 mg/mL) was added to each well, cultured for a further 4 h. Subsequently, 150 µL of dimethyl sulfoxide was added to the well after removing the medium. An ELISA plate reader (Biotek, Winooski, VT, USA) was used to measure the optical density (OD) value at 570 nm wavelength. Additionally, the half maximal inhibitory concentration (IC_50_) value was determined by GraphPad Prism 5 software. The cell inhibition rate was calculated as follows: Cell inhibition rate (%) = (1 − OD treated/OD control) × 100%
where OD treated is the cells obtained by nanoparticle treatment and OD control is the cells obtained by medium treatment.

### 3.12. Statistical Analysis

The experimental data were expressed as mean ± standard deviation (>3 independent experiments). The data between groups were compared by one-way analysis of variance (ANOVA) (SPSS 25.0, SPSS Company, Chicago, IL, USA), and *p* < 0.05 indicated statistical significance.

## 4. Conclusions

The dendritic amphiphilic hybrid compound G2-C18 can be dissolved in deionized water and self-assembled into nanoparticles (NPs). Using celastrol (CSL) as the model drug, CSL-G2-C18 NPs were prepared via the antisolvent precipitation method. Meanwhile, in order to preliminarily explore the effect of lipophilic segment aliphatic chain C18 on the antitumor activity of NPs, CSL-G2 NPs were prepared under the same conditions using G2 as the control group. After successful preparation, the two CSL NPs presented similar drug-loading contents, particle sizes, polydispersity indexes, physical states, and stabilities. However, based on the different hydrophobic interactions, these two CSL NPs showed different morphologies and release rates. CSL-G2 NPs exhibited a spherical morphology and sustained release for four days. CSL-G2-C18 NPs exhibited a vesicular morphology, and their release process was prolonged to six days. Moreover, different morphologies resulted in different antitumor activities. Compared with CSL-G2 NPs, the IC50 of CSL-G2-C18 NPs was reduced two-fold. Thus, CSL-G2-C18 NPs showed higher antitumor efficacy. These results suggest that the aliphatic chain C18 in the amphiphilic compound G2-C18 could affect the antitumor efficacy of a nano drug delivery system effectively, which provides a reference for the design of future NPs.

## Figures and Tables

**Figure 1 molecules-28-00069-f001:**
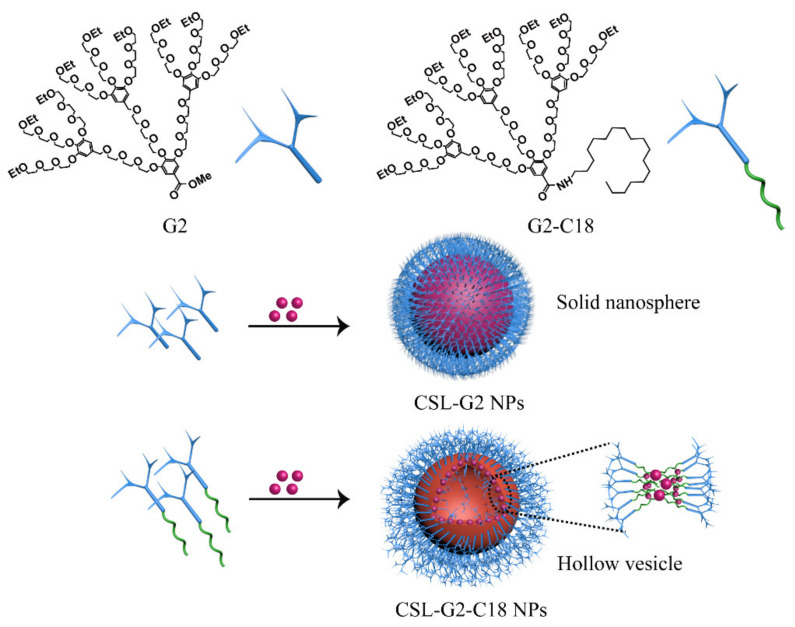
Chemical structure of nanocarriers G2 and G2-C18, CSL NPs cartoon illustration.

**Figure 2 molecules-28-00069-f002:**
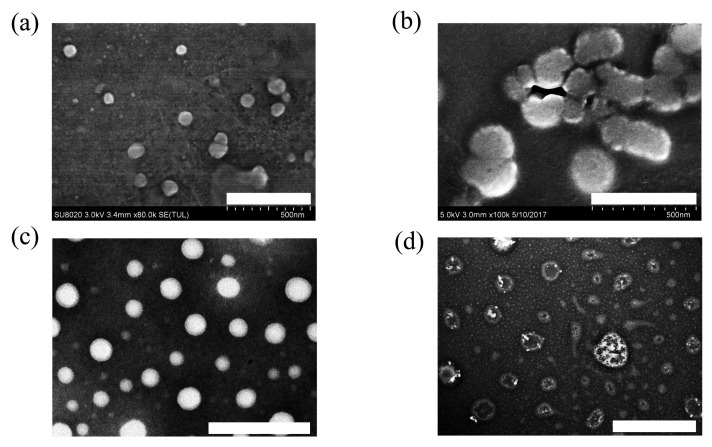
TEM images of G2 (**a**) (scale bar 500 nm), G2-C18 (**b**) (scale bar 500 nm), CSL-G2 NPs (**c**) (scale bar 200 nm), and CSL-G2-C18 NPs (**d**) (scale bar 500 nm).

**Figure 3 molecules-28-00069-f003:**
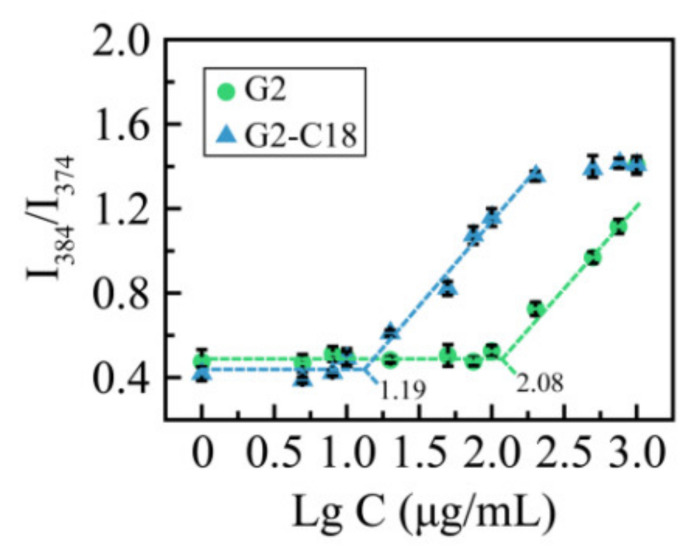
Intensity ratio I_384_/I_374_ curves as a function of G2 and G2-C18 concentration.

**Figure 4 molecules-28-00069-f004:**
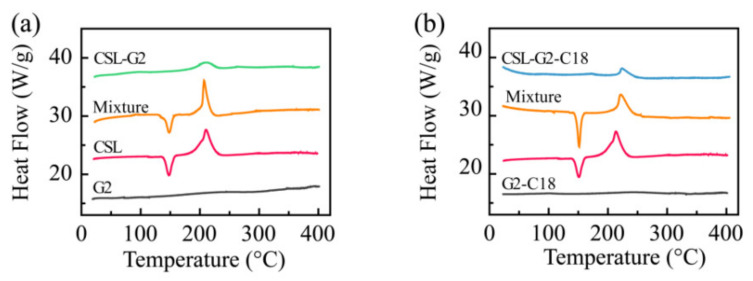
DSC thermograms of CSL, nanocarriers, CSL-loaded NPs, and the physical mixture of CSL and nanocarriers: G2 series (**a**) and G2-C18 series (**b**).

**Figure 5 molecules-28-00069-f005:**
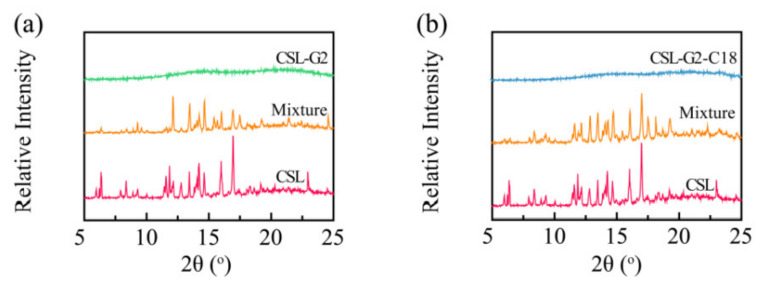
XRD patterns of CSL, CSL-loaded NPs, and the physical mixture of CSL and nanocarriers: G2 series (**a**) and G2-C18 series (**b**).

**Figure 6 molecules-28-00069-f006:**
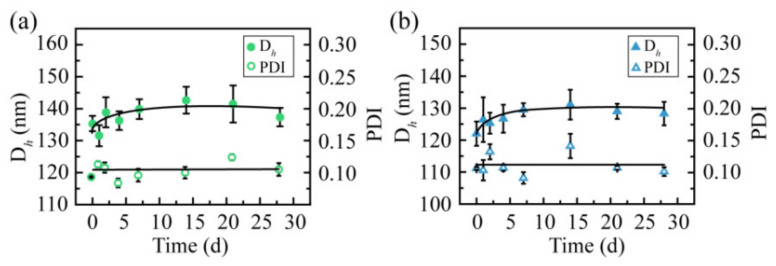
Particle size and polydispersity index of CSL-loaded nanoparticles during storage at 4 °C: CSL-G2 NPs (**a**) and CSL-G2-C18 NPs (**b**), *n* = 3.

**Figure 7 molecules-28-00069-f007:**
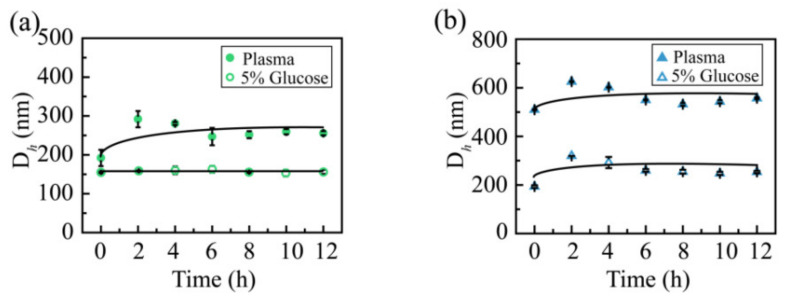
Particle size of the CSL-G2 NPs (**a**) and CSL-G2-C18 NPs (**b**) in glucose solution (5%) and plasma.

**Figure 8 molecules-28-00069-f008:**
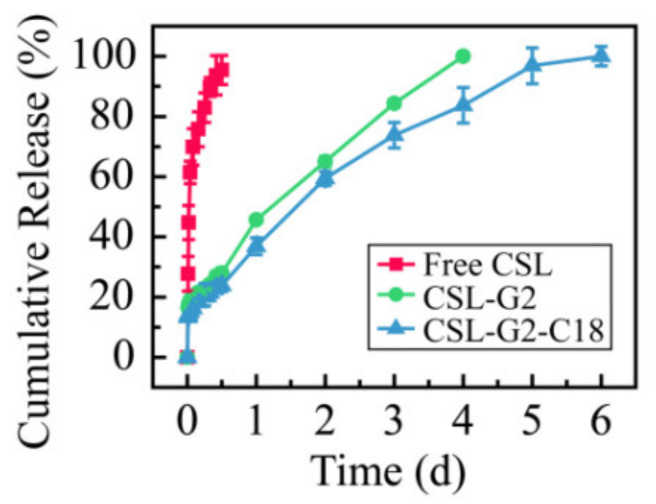
Cumulative release rate of CSL in 5% glucose solution at 37 °C over 144 h.

**Figure 9 molecules-28-00069-f009:**
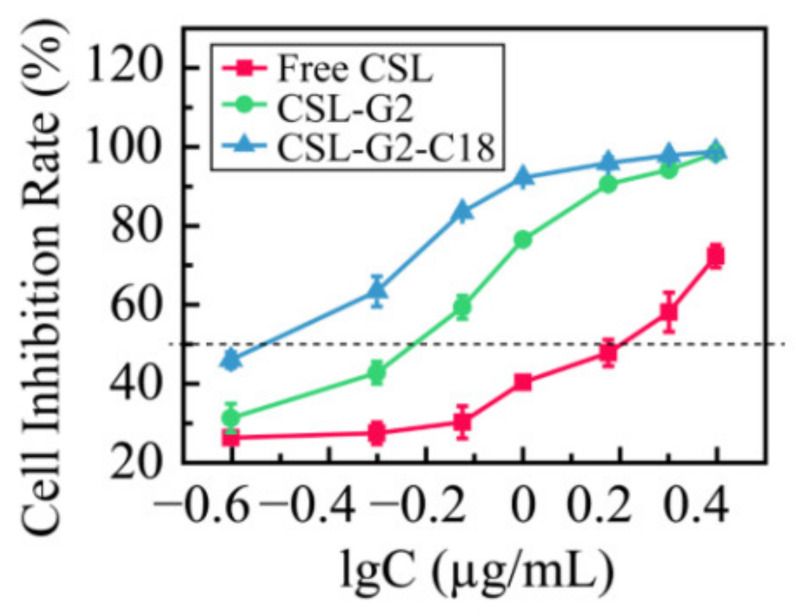
Cytotoxicity of CSL NPs to 4T1 cells after 48 h incubation (*n* = 5).

## Data Availability

Not applicable.

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
