# Peer review of "Effect of Lipophilic Chains on the Antitumor Effect of a Dendritic Nano Drug Delivery System"

_molecules, 2022, doi:10.3390/molecules28010069_

Round 1
Reviewer 1 Report
The manuscript from Guo and co-workers reports synthesis of an amphiphilic hybrid dendritic molecules containing hydrophilic G2 dendron and hydrophobic C18 chain. It can form “core-shell” nanoparticles loaded with celastrol (CSL) via antisolvent precipitation method. The interactions between C18 chain and hydrophobic CSL is studied in detail. They also carried out a variety of measurements to investigate properties of the nanoparticles, and displayed the data properly. Overall, this work is interesting and has the potential to be translated from basic research to clinical implications. The manuscript is generally well organized, I thus would recommend it to be accepted for publication. But there are two notes “Error! Reference source not found” in page 6, line 209, and page 7, line 225. The authors should carefully check them and make corrections.
Reviewer 2 Report
Review on Manuscript molecules-2099440
The article is about the preparation of drug delivery particles using oligoethylene glycol dendron (G2).
My comments:
- The topic and timeliness of the article is appropriate, but the figures drawn from the results are quite small, making it difficult to interpret the results in this way. I would strongly recommend enlarging the figures by at least 50%.
- The size range of nanoparticles (also based on the IUPAC designation) is less than ~100 nm, so I would call particles of ~ 140 nm “colloidal particles” rather than nanoparticles. Colloidal particles (1-500 nm).
- line 82. and line 225 Reference error, please modify.
- line 175. “XRD curves”. please use “XRD diffractogram” or “XRD patterns” naming.
- Figure 5. 5-25 degree is max. enough to present on x-axes.
- Line 280. for determination of the zeta potential values what kind of equation was used? Given the size of the particles, it makes a difference which one you count.
Major revision is required.
Round 2
Reviewer 2 Report
I accept the corrected version